# Comparison of the intrageneric neutralization scope of monospecific, bispecific/monogeneric and polyspecific/monogeneric antisera raised in horses immunized with sub-Saharan African snake venoms

**Andrés Sánchez, Gina Durán, Álvaro Segura, María Herrera, Mariángela Vargas, Mauren Villalta, Mauricio Arguedas, Edwin Moscoso, Deibid Umaña, Aarón Gómez, José María Gutiérrez, Guillermo León**  *

Instituto Clodomiro Picado, Facultad de Microbiología, Universidad de Costa Rica, San José, Costa Rica

* guillermo.leon@ucr.ac.cr

## Abstract

### Background

Snakebite envenomation inflicts a high burden of mortality and morbidity in sub-Saharan Africa. Antivenoms are the mainstay in the therapy of envenomation, and there is an urgent need to develop antivenoms of broad neutralizing efficacy for this region. The venoms used as immunogens to manufacture snake antivenoms are normally selected considering their medical importance and availability. Additionally, their ability to induce antibody responses with high neutralizing capability should be considered, an issue that involves the immunization scheme and the animal species being immunized.

### Methodology/Principal findings

Using the lethality neutralization assay in mice, we compared the intrageneric neutralization scope of antisera generated by immunization of horses with monospecific, bispecific/monogeneric, and polyspecific/monogeneric immunogens formulated with venoms of *Bitis* spp., *Echis* spp., *Dendroaspis* spp., spitting *Naja* spp. or non-spitting *Naja* spp. It was found that the antisera raised by all the immunogens were able to neutralize the homologous venoms and, with a single exception, the heterologous congeneric venoms (considering spitting and non-spitting *Naja* separately). In general, the polyspecific antisera of *Bitis* spp, *Echis* spp, and *Dendroaspis* spp gave the best neutralization profile against venoms of these genera. For spitting *Naja* venoms, there were no significant differences in the neutralizing ability between monospecific, bispecific and polyspecific antisera. A similar result was obtained in the case of non-spitting *Naja* venoms, except that polyspecific antiserum was more effective against the venoms of *N. melanoleuca* and *N. nivea* as compared to the monospecific antiserum.

**Data Availability Statement:** All relevant data are in the manuscript and its supporting information files.

**Funding:** This study was supported by a Wellcome Trust grant [Reference 220517/Z/20/Z] awarded to GL and JMG, and by Vicerrectoría de Investigación, Universidad de Costa Rica [projects 741-A0-804 and 741-C0-523] awarded to GL. The funders had no role in study design, data collection and analysis, decision to publish, or preparation of the manuscript. All the authors received salary from Universidad de Costa Rica.

**Competing interests:** I have read the journal's policy and the authors of this manuscript have the following competing interests: The authors work at Instituto Clodomiro Picado, where an antivenom for use in sub-Saharan Africa is manufactured.

## Conclusions/Significance

The use of polyspecific immunogens is the best alternative to produce monogeneric antivenoms with wide neutralizing coverage against venoms of sub-Saharan African snakes of the *Bitis*, *Echis*, *Naja* (non-spitting) and *Dendroaspis* genera. On the other hand, a monospecific immunogen composed of venom of *Naja nigricollis* is suitable to produce a monogeneric antivenom with wide neutralizing coverage against venoms of spitting *Naja* spp. These findings can be used in the design of antivenoms of wide neutralizing scope for sub-Saharan Africa.

### Author summary

Parenteral administration of antivenoms is the core of the current treatment of snakebite envenomations, and there is an urgent need to produce antivenoms of wide neutralizing efficacy for sub-Saharan Africa. The active substance of antivenoms are antibodies (or antibody fragments) purified from plasma of horses or sheep immunized by the repeated injection of snake venoms. Generally, these antibodies can neutralize the venoms used as immunogens and other related venoms. Normally, the venoms used as immunogens are selected considering their medical importance and availability. To complement these criteria with information regarding the immunogenicity of venoms, we compared monospecific, bispecific/monogeneric, and polyspecific/monogeneric antisera towards venoms of *Bitis* spp., *Echis* spp., *Dendroaspis* spp., spitting *Naja* spp. or non-spitting *Naja* spp, regarding their intrageneric neutralization scope, evaluated by the lethality neutralization assay in mice. We found that the polyspecific antisera against venoms of *Bitis* spp, *Echis* spp, *Dendroaspis* spp, or non-spitting *Naja* gave the best neutralization profile. On the other hand, the monospecific, bispecific and polyspecific antisera towards venoms of spitting *Naja* venoms showed a similar performance. This information suggests that polyspecific immunogens could be the best alternative to produce antivenoms with the widest neutralizing coverage against sub-Saharan African snake venoms.

## Introduction

Snakebite envenomation affects thousands of people every year in sub-Saharan Africa, causing death and disability, especially in impoverished rural communities [1]. The mainstay in the therapy of these envenomations is the timely administration of safe and effective antivenoms [2]. Snake antivenoms are formulations of whole immunoglobulins, F(ab')$_2$ or Fab fragments purified from the plasma of animals (e.g., horses or sheep) immunized with snake venoms [3]. The traditional immunization procedure consists of the periodic injection of variable amounts of one or several venoms, mixed with immunological adjuvants that enhance the antibody response of the animals [4].

Venoms used as immunogens are normally selected according to the medical importance of the snakes in the regions where the antivenom is intended to be used. According to the World Health Organization (WHO), the sub-Saharan African snakes with major potential to induce envenomations of high incidence and severity are those of the *Bitis* (puff-adders), *Echis* (saw-scale/carpet vipers), *Dendroaspis* (mambas) and *Naja* (spitting and non-spitting *Naja*)

genera [3]. Together, these snakes induce thousands of envenomations, deaths and sequelae, mostly affecting impoverished rural communities in sub-Saharan Africa [1].

Envenomation by African snakes can be classified in three different syndromes, depending on the species involved: 1) Marked local swelling with coagulable blood (*i.e.*, pain, progressive swelling, blisters, necrosis, hypotension and eventually shock) produced by *Bitis* spp. and spitting *Naja* spp.; 2) Marked local swelling with incoagulable blood and/or spontaneous systemic bleeding, predominantly caused by *Echis* spp.; and 3) Neurotoxicity (*i.e.*, ptosis, diplopia, dysphagia, and the later progression of muscular paralysis to affect the respiratory muscles and eventually produce respiratory arrest and death), characteristic of envenomations by *Dendroaspis* spp. and non-spitting *Naja* spp. [5].

The clinical characteristics of envenomations are related to the composition of the venoms. The main toxins in the venoms of *Bitis* spp. and *Echis* spp. are hemorrhagic and/or procoagulant $Zn^{2+}$-dependent metalloproteinases (SVMPs), and necrotizing phospholipases $A_2$ ($PLA_2$s) [6, 7]. In the case of *Dendroaspis* spp. and *Naja* spp., neurotoxic Kunitz-type serine proteinase inhibitor-like toxins (KUNs), neurotoxic and cytotoxic three-finger toxins (3FTxs), and $PLA_2$s are the main families responsible for toxicity [8–10]. These toxins have physicochemical characteristics that confer them with toxicity and immunogenicity. In general, SVMPs are highly immunogenic, while KUN, 3FTxs and $PLA_2$s induce weak antibody responses [4].

In addition to medical importance and immunogenicity, the selection of venoms used as immunogens should consider the neutralization scope of the antibody response induced in particular animal species by specific immunization strategies. According to our previous results in a rabbit model, the venoms inducing antibody responses with the broadest intrageneric neutralization and immunorecognition scope are those of *Bitis gabonica* and *B. rhinoceros*; *Echis leucogaster*; *Dendroaspis jamesoni* and *D. viridis*; *Naja nigricollis* and *N. ashei*, for spitting *Naja*; and *N. senegalensis* and *N. haje*, for non-spitting *Naja* [11–13]. These previous results, related to the intrageneric cross-reactivity of antisera raised against venoms of the medically most important sub-Saharan African snakes, provided valuable information for the experimental design of the present work, and hence for the scaling up of hyperimmune plasma production in horses.

In this work, we immunized horses with monospecific, bispecific/monogeneric and polyspecific/monogeneric mixtures of venoms of *Bitis* spp., *Echis* spp., *Dendroaspis* spp., spitting *Naja* spp. or non-spitting *Naja* spp., and compared their antibody responses regarding its intrageneric neutralization of lethality in mice, to determine the composition of the immunogens that stimulate the antibody responses with the broadest neutralizing scope within each genus. The results of this work contribute to the rational, evidence-based design of immunization strategies that could result in more effective antivenoms for the treatment of snakebite envenomations in sub-Saharan Africa.

## Materials and methods

### Ethics

All procedures used in this study were approved by the Institutional Committee for the Care and Use of Laboratory Animals (CICUA) of Universidad de Costa Rica (Proceedings 82–08 and 39–20) and meet the International Guiding Principles for Biomedical Research Involving Animals [14].

### Animal management

Horses were maintained in a farm located at 1495 m above sea level, with access to water and pasture *ad libitum*, at a population density of 2 horses/Ha. The grazing technique was in

paddocks planted with "star" grass. The diet was supplemented with pelleted feed enriched with proteins, vitamins, and minerals. Before starting the immunization, the horses completed a two-month quarantine during which they were dewormed, acclimatized, and brought to optimal physical condition. Mice were obtained from the Bioterium of Instituto Clodomiro Picado and handled in Tecniplast Eurostandard Type II 1264C cages (L25.0 x W40.0 x H 14.0 cm), five mice per cage, at 18–24˚C, 60–65% relative humidity, and 12:12 light-dark cycle, with food and water *ad libitum*.

## Venoms

Venoms of adult specimens of *Bitis arietans* (unspecified origin, batch #322.061), *B. gabonica* (unspecified origin, batch #725.031), *B. nasicornis* (unspecified origin, batch #500.102), *B. rhinoceros* (from Ghana, batch #701.070), *Echis leucogaster* (from Mali, batch #623.070), *E. ocellatus* (unspecified origin, batch #216.031), *E. pyramidum* (from Egypt, batch #523.070), *Dendroaspis angusticeps* (Tanzania, Mozambique; batch #305.000), *D. jamesoni* (Cameroon; batch #923.011), *D. polylepis* (unknown origin; batch #416.031) and *D. viridis* (Ghana, Togo; batch #516.001), *Naja anchietae* (Namibia, batch #527.002), *N. annulifera* (Mozambique, batch #622.040), *N. ashei* (Kenya, batch #410.191), *N. haje* (unknown origin, batch #222.061), *N. katiensis* (Burkina Faso, batch #705.010), *N. melanoleuca* (unknown origin, batch #516.031), *N. mossambica* (Tanzania, batch #627.002), *N. nigricincta* (South Africa, batch #507.081), *N. nigricollis* (unknown origin, batch #616.031), *N. nivea* (South Africa, batch #524.010) and *N. senegalensis* (Mali, batch #805.010) were purchased from Latoxan (Portesdès Valence, France). Freeze-dried venoms were obtained from the supplier and stored at -40˚C until use. Solutions of venoms were prepared immediately before use.

## Immunization of horses

Five groups of four creole horses (250–400 kg body weight) were immunized towards venoms of one, two or several snake species of the same genera (i.e., *Bitis* spp., *Echis* spp., *Dendroaspis* spp., spitting *Naja* spp., or non-spitting *Naja* spp). The immunization protocol is described in Fig 1 and Table 1.

In the first immunization stage, horses were immunized with venoms of single species to produce monospecific sera. In the second stage, the same horses were immunized with a mixture of equal parts of two venoms to produce bispecific/monogeneric sera. In the third stage, horses were immunized with mixtures of equal parts of several venoms to produce polyspecific/monogeneric sera. The selection of the venoms to be used for generating the monospecific antisera derives from previous studies using a rabbit model of immunization [11–13]. In the first two boosters of each stage, venoms were emulsified in Montanide ISA 50V2 (1 mL total volume of each injection, 0.5 mL Montanide and 0.5 mL of venom dissolved in sterile saline solution, i.e., 0.15 M NaCl, pH 7.2). In the other boosters, venoms were dissolved in 2 mL sterile saline solution. Montanide ISA 50V2 was used as adjuvant owing to its ability to enhance the antibody response of horses immunized towards the venoms of African snakes, even though it induces some adverse effects at the injection site. This adjuvant is composed of an injectable mineral oil, vegetable oleic acid and anhydro mannitol ether octodecenoate as emulsifier [15]. The immunogens were administered in a single injection site in the back of the horse by the subcutaneous (SC) route. Before the onset of immunization, and fifteen days after each immunogen injection, 10 mL-blood samples were collected from the jugular vein to monitor hematological and serum chemistry parameters. The samples collected in weeks 13, 25 and 37 (Fig 1 and Table 1) were pooled per group and used to determine the anti-lethal $ED_{50}$

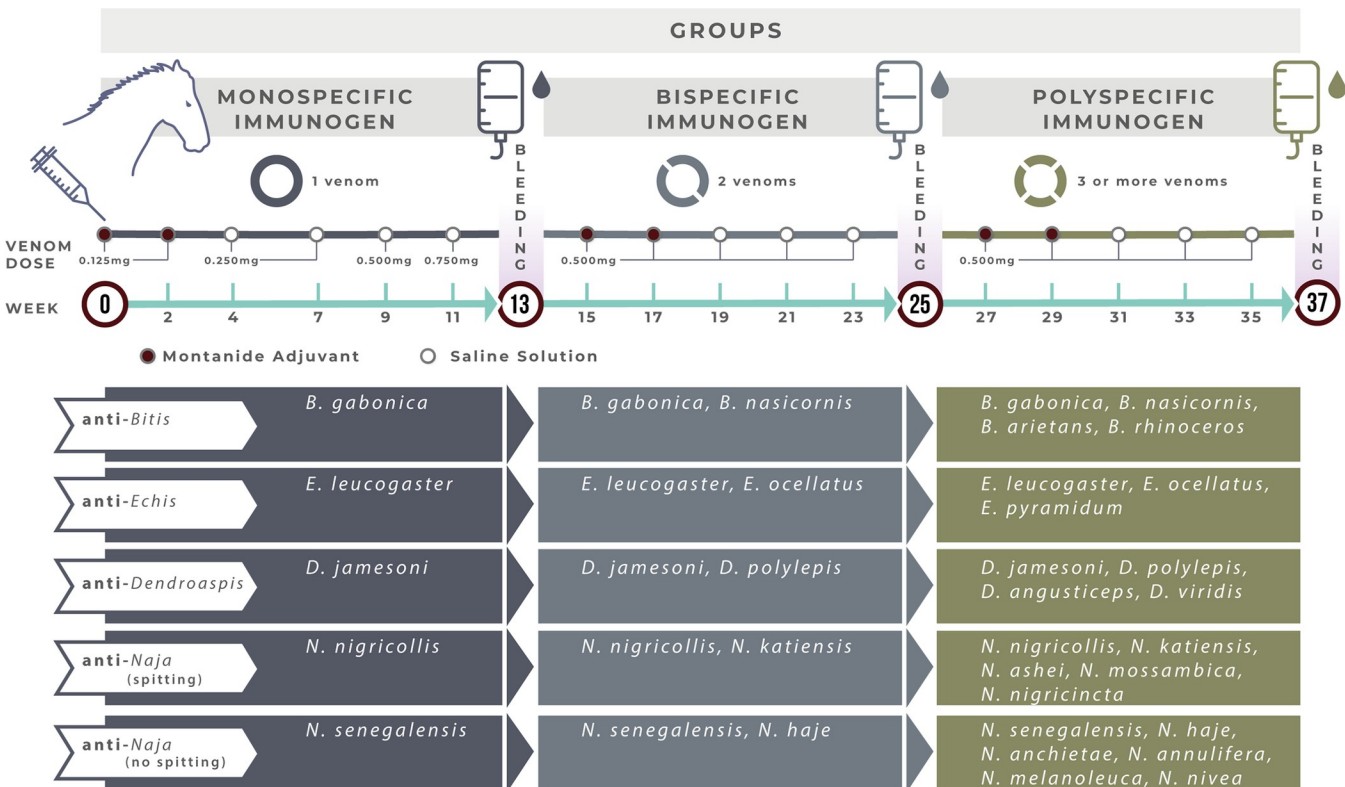

**Fig 1. Immunization scheme of horses with sub-Saharan African snake venoms to produce monospecific, bispecific/monogeneric and polyspecific/monogeneric antisera.** Groups of four creole horses were immunized to produce fifteen monogeneric antisera towards venoms of *Bitis* spp., *Echis* spp., *Dendroaspis* spp., spitting *Naja* spp., or non-spitting *Naja* spp snakes. The immunogens were injected by the subcutaneous route, in a single site, in the back of the horses [4].

of the antisera in a mouse model. The clinical and physical status of the animals were constantly under veterinary supervision.

On the other hand, an industrial polygeneric plasma was collected from 25 horses, which during the last three years have been periodically immunized with the venoms of *B. arietans*, *E. ocellatus*, *D. polylepis* and *N. nigricollis*. This antivenom is prepared using the initial immunization scheme described by Gutiérrez and co-workers [16]. After that, horses have been regularly boosted, every two months, with the same venom mixture, and bled ten days after each booster immunization for the production of EchiTAb-plus-ICP antivenom. Comparisons with this plasma were done in order to project the usefulness of the experimental data generated in this study in the industrial production of snake antivenoms.

## Body condition, hematological and serum chemistry analyses

Body condition score (BCS) of horses was evaluated by tactile exploration and visual assessment of the fat accumulation in anatomical sites, such as behind the shoulders, over the ribs, along the neck, along the withers, the crease down back and the tailhead [17]. Scores range from 1 (extremely emaciated) to 9 (extremely fat). Lesions developed at the injection site were evaluated and classified based on their severity, according to Arguedas et al. (2022), in the following categories: Category 1 (mild painful lesions with diffuse edema around the injection site), Category 2 (well-circumscribed lesions with soft abscesses which eventually develop fistula), or Category 3 (well circumscribed or diffused lesions associated with the development of solid fibrous tissue)

**Table 1. Immunization of horses to produce monospecific, bispecific and polyspecific antisera against the venoms of the medically most relevant snakes from sub-Saharan Africa.**

| Week | Groups of horses | | | | |
|---|---|---|---|---|---|
| | anti-*Bitis* | anti-*Echis* | anti-*Dendroaspis* | anti-*Naja* (spitting) | anti-*Naja* (no spitting) |
| 0 | Bga (0.125 mg) Montanide | Ele (0.125 mg) Montanide | Dja (0.125 mg) Montanide | Nng (0.125 mg) Montanide | Nse (0.125 mg) Montanide |
| 2 | Bga (0.125 mg) Montanide | Ele (0.125 mg) Montanide | Dja (0.125 mg) Montanide | Nng (0.125 mg) Montanide | Nse (0.125 mg) Montanide |
| 4 | Bga (0.250 mg) Saline solution | Ele (0.250 mg) Saline solution | Dja (0.250 mg) Saline solution | Nng (0.250 mg) Saline solution | Nse (0.250 mg) Saline solution |
| 7 | Bga (0.250 mg) Saline solution | Ele (0.250 mg) Saline solution | Dja (0.250 mg) Saline solution | Nng (0.250 mg) Saline solution | Nse (0.250 mg) Saline solution |
| 9 | Bga (0.500 mg) Saline solution | Ele (0.500 mg) Saline solution | Dja (0.500 mg) Saline solution | Nng (0.500 mg) Saline solution | Nse (0.500 mg) Saline solution |
| 11 | Bga (0.750 mg) Saline solution | Ele (0.750 mg) Saline solution | Dja (0.750 mg) Saline solution | Nng (0.750 mg) Saline solution | Nse (0.750 mg) Saline solution |
| 13 | Bleeding | Bleeding | Bleeding | Bleeding | Bleeding |
| 15 | Bga+Bna (0.500 mg) Montanide | Ele+Eoc (0.500 mg) Montanide | Dja+Dpo (0.500 mg) Montanide | Nng+Nka (0.500 mg) Montanide | Nse+Nha (0.500 mg) Montanide |
| 17 | Bga+Bna (0.500 mg) Montanide | Ele+Eoc (0.500 mg) Montanide | Dja+Dpo (0.500 mg) Montanide | Nng+Nka (0.500 mg) Montanide | Nse+Nha (0.500 mg) Montanide |
| 19 | Bga+Bna (0.500 mg) Saline solution | Ele+Eoc (0.500 mg) Saline solution | Dja+Dpo (0.500 mg) Saline solution | Nng+Nka (0.500 mg) Saline solution | Nse+Nha (0.500 mg) Saline solution |
| 21 | Bga+Bna (0.500 mg) Saline solution | Ele+Eoc (0.500 mg) Saline solution | Dja+Dpo (0.500 mg) Saline solution | Nng+Nka (0.500 mg) Saline solution | Nse+Nha (0.500 mg) Saline solution |
| 23 | Bga+Bna (0.500 mg) Saline solution | Ele+Eoc (0.500 mg) Saline solution | Dja+Dpo (0.500 mg) Saline solution | Nng+Nka (0.500 mg) Saline solution | Nse+Nha (0.500 mg) Saline solution |
| 25 | Bleeding | Bleeding | Bleeding | Bleeding | Bleeding |
| 27 | Bga+Bna+Bar+Bri (0.500 mg) Montanide | Ele+Eoc+Epy (0.500 mg) Montanide | Dja+Dpo+Dan+Dvi (0.500 mg) Montanide | Nng+Nka+Nsh +Nmo+Nta (0.500 mg) Montanide | Nse+Nha+Nch +Nan+Nme+Nnv (0.500 mg) Montanide |
| 29 | Bga+Bna+Bar+Bri (0.500 mg) Montanide | Ele+Eoc+Epy (0.500 mg) Montanide | Dja+Dpo+Dan+Dvi (0.500 mg) Montanide | Nng+Nka+Nsh +Nmo+Nta (0.500 mg) Montanide | Nse+Nha+Nch +Nan+Nme+Nnv (0.500 mg) Montanide |
| 31 | Bga+Bna+Bar+Bri (0.500 mg) Saline solution | Ele+Eoc+Epy (0.500 mg) Saline solution | Dja+Dpo+Dan+Dvi (0.500 mg) Saline solution | Nng+Nka+Nsh +Nmo+Nta (0.500 mg) Saline solution | Nse+Nha+Nch +Nan+Nme+Nnv (0.500 mg) Saline solution |
| 33 | Bga+Bna+Bar+Bri (0.500 mg) Saline solution | Ele+Eoc+Epy (0.500 mg) Saline solution | Dja+Dpo+Dan+Dvi (0.500 mg) Saline solution | Nng+Nka+Nsh +Nmo+Nta (0.500 mg) Saline solution | Nse+Nha+Nch +Nan+Nme+Nnv (0.500 mg) Saline solution |

(*Continued*)

**Table 1.** (Continued)

| Week | Groups of horses | | | | |
|---|---|---|---|---|---|
| | anti-*Bitis* | anti-*Echis* | anti-*Dendroaspis* | anti-*Naja* (spitting) | anti-*Naja* (no spitting) |
| 35 | Bga+Bna+Bar+Bri (0.500 mg) Saline solution | Ele+Eoc+Epy (0.500 mg) Saline solution | Dja+Dpo+Dan+Dvi (0.500 mg) Saline solution | Nng+Nka+Nsh +Nmo+Nta (0.500 mg) Saline solution | Nse+Nha+Nch +Nan+Nme+Nnv (0.500 mg) Saline solution |
| 37 | Bleeding | Bleeding | Bleeding | Bleeding | Bleeding |

Venoms used were: *Bitis gabonica* (Bga), *B. nasicornis* (Bna), *B. arietans* (Bar), *B. rhinoceros* (Bri), *Echis leucogaster* (Ele), *E. ocellatus* (Eoc), *E. pyramidum* (Epy), *Dendroaspis jamesoni* (Dja), *D. polylepis* (Dpo), *D. angusticeps* (Dan), *D. viridis* (Dvi), *Naja nigricollis* (Nng), *N. katiensis* (Nka), *N. ashei* (Nsh), *N. mossambica* (Nmo), *N. nigricincta* (Nta), *N. senegalensis* (Nse), *N. haje* (Nha), *N. anchietae* (Nch), *N. annulifera* (Nan), *N. melanoleuca* (Nme), *N. nivea* (Nnv).

[15]. Hematological analyses (i.e., hematocrit and hemoglobin concentration) were carried out in a Veterinary Hematology Analyzer (Exigo Eos Hematology System; Boule Diagnostics AB, Stockholm, Sweden). Plasma chemistry analyses were carried out in a clinical chemistry analyzer (Spin200E Automatic biochemistry analyzer; Spinreact, Barcelona, Spain). Creatine kinase (CK) was determined by the corresponding International Federation of Clinical Chemistry and Laboratory Medicine (IFCC) method. Creatinine was determined by a kinetic modification of the Jaffe colorimetric method [18] and serum urea by a modification of the Talke and Schubert method [19]. Aspartate transaminase (AST) and alkaline phosphatase (ALP) were determined by the corresponding IFCC methods. Gamma-glutamyl transferase (GGT) was determined by a modification of the Szasz procedure [20]. Total protein concentration was determined by the Biuret method [21], albumin concentration by the bromocresol green colorimetric method [22], and the gamma gap was determined as the difference between total protein and albumin concentration.

## Determination of median effective dose ($ED_{50}$) of antisera

The efficacy of antisera to neutralize lethality of venoms was assessed in a mouse model. Mixtures of a constant amount of venom and variable dilutions of serum or heat inactivated plasma were prepared and incubated at 37°C during 30 min. Then, 0.5 mL of each mixture, containing a challenge dose of 2 median lethal doses ($LD_{50}$) of venom, were injected by the intraperitoneal route (IP) in groups of five CD-1 mice (16–18 g, both sexes, and randomly allocated). Fifteen min before injection, the mice were pretreated with the analgesic tramadol, administered by the subcutaneous (SC) route, at a dose of 50 mg/kg [23]. Control mice received the same dose of venom with no serum. Instead of the characteristic 3–6 $LD_{50}$s, the 2 $LD_{50}$s dose was selected as challenge dose to increase the sensitivity of the assay, thus favoring the detection of cross-reactivity of antisera against venoms. Challenge doses were defined based on $LD_{50}$ values reported in recently published works [11–13], which correspond to the same batches of the venoms hereby used (Table 2). The number of deaths in each group was recorded at 6 h after injection and used to calculate the median effective dose (i.e., $ED_{50}$: the ratio mg venom/mL serum in which 50% of the challenged mice survive) and the corresponding 95% confidence interval (CI), by Probits [24, 25]. Experimenters were not blinded to the identity of the samples.

## Statistical analyses

The significance of the differences between $ED_{50}$ values was assessed based on the overlap of the 95% CI. Values in which 95% CI do not overlap are considered significantly different.

**Table 2. Median lethal dose (LD$_{50}$) of venoms, determined by the intraperitoneal route, expressed as µg venom per mouse. Values in parentheses represent the 95% confidence intervals (CI)***.

| Venom | LD$_{50}$ (95% CI) | Venom | LD$_{50}$ (95% CI) |
|---|---|---|---|
| *Bitis arietans* | 22.0 (12.9–31.6) | *Naja anchietae* | 69.7 (47.3–95.6) |
| *Bitis gabonica* | 29.4 (22.1–38.5) | *Naja annulifera* | 63.5 (48.5–84.6) |
| *Bitis nasicornis* | 47.4 (33.2–67.9) | *Naja ashei* | 21.7 (15.8–30.5) |
| *Bitis rhinoceros* | 31.7 (24.9–42.0) | *Naja haje* | 1.9 (1.4–2.7) |
| *Echis leucogaster* | 45.5 (33.0–62.0) | *Naja katiensis* | 18.9 (13.7–24.8) |
| *Echis ocellatus* | 31.2 (21.1–49.1) | *Naja melanoleuca* | 6.4 (4.9–8.9) |
| *Echis pyramidum* | 39.2 (26.4–52.2) | *Naja mossambica* | 23.0 (16.1–32.6) |
| *Dendroaspis angusticeps* | 21.7 (15.8–30.5) | *Naja nigricincta* | 21.7 (15.8–30.5) |
| *Dendroaspis jamesoni* | 18.5 (14.2–26.5) | *Naja nigricollis* | 18.4 (10.7–26.4) |
| *Dendroaspis polylepis* | 6.6 (4.9–8.6) | *Naja nivea* | 22.4 (17.8–28.3) |
| *Dendroaspis viridis* | 7.3 (5.5–12.6) | *Naja senegalensis* | 7.4 (5.7–10.1) |

*Values of *Bitis* spp. and *Echis* spp. venoms were from Gómez et al. (2022) [11]. Values of *Naja* spp. venoms from Gómez et al. (2023) [12], and *Dendroaspis* spp. venoms from Gómez et al. (2024) [13].

## Results and discussion

### Horse welfare

Only 3.1% of venom immunogen injections produced local tissue lesions in horses. Nine local lesions were recorded in horses after 120 injections applied during the immunization with monospecific immunogens: one in the anti-*Bitis* group, five in the anti-spitting *Naja* group and three in the anti non-spitting *Naja* group. All these lesions were produced by the immunogens emulsified in Montanide, applied during the first and second weeks of immunization. Such lesions corresponded to Category 2 in the classification proposed by Arguedas et al. (2022) [15] (i.e., well-circumscribed lesions characterized by the formation of soft abscesses which eventually ulcerate and/or open a fistula through which a bloody pus-like material is discharged). Only one out of 100 injections of bispecific immunogens resulted in a local lesion that corresponded to Category 3 of the Arguedas et al. classification (i.e., well circumscribed, or diffused lesions characterized by the development of solid fibrous tissue, which may or may not include fistula formation) [15]. This lesion was observed in a horse of the anti-*Bitis* group, as a consequence of the immunogen emulsified in Montanide, injected in the fifteenth week. No lesions were observed in horses as a consequence of any of the 100 injections of the polyspecific immunogens. Generally, the injuries healed on their own without major complications. The lesions were treated by washing with soap and water, and the topical application of a 2% iodine solution and an antiseptic/healing spray (Bactrovet silver AM, Laboratorios König S.A.).

During the entire experiment, the horses did not show weight loss or a decrease in their body condition score (initial value of 2.0–4.0). A mild drop of the hematocrit and hemoglobin

values was found in the horses immunized with venoms of *Bitis* spp and *Naja* spp (S1 Table). Moreover, an increment in CK values, as compared to reference values, was found in the plasma of horses immunized towards venoms of *Echis* spp, *Dendroaspis* spp and *Naja* spp (S1 Table), which could be related to the myotoxic effects of these venoms in the injection site. Values of creatinine and urea were within the normal range, indicating no renal affectation (S1 Table), while values of AST, ALP and GGT were also within the normal range, thus evidencing the absence of hepatic problems (S1 Table). Finally, plasma concentration of total protein, albumin and globulins were also within the normal ranges (S1 Table). Taken together, these results underscore the absence of systemic toxicity during the immunization schemes used.

## Neutralization of lethality by antisera raised against *Bitis* spp venoms

The monospecific anti-*Bitis* serum was prepared by mixing the sera of four horses immunized with *B. gabonica* venom. This venom was selected on the basis of previous findings in a rabbit model [11]. The antiserum generated neutralized not only the lethality induced by the homologous venom, but it also cross-neutralized the lethality induced by the venoms of the other *Bitis* species (i.e., heterologous venoms of *B. arietans*, *B. nasicornis* and *B. rhinoceros*; Fig 2), with variations in the $ED_{50}$ values.

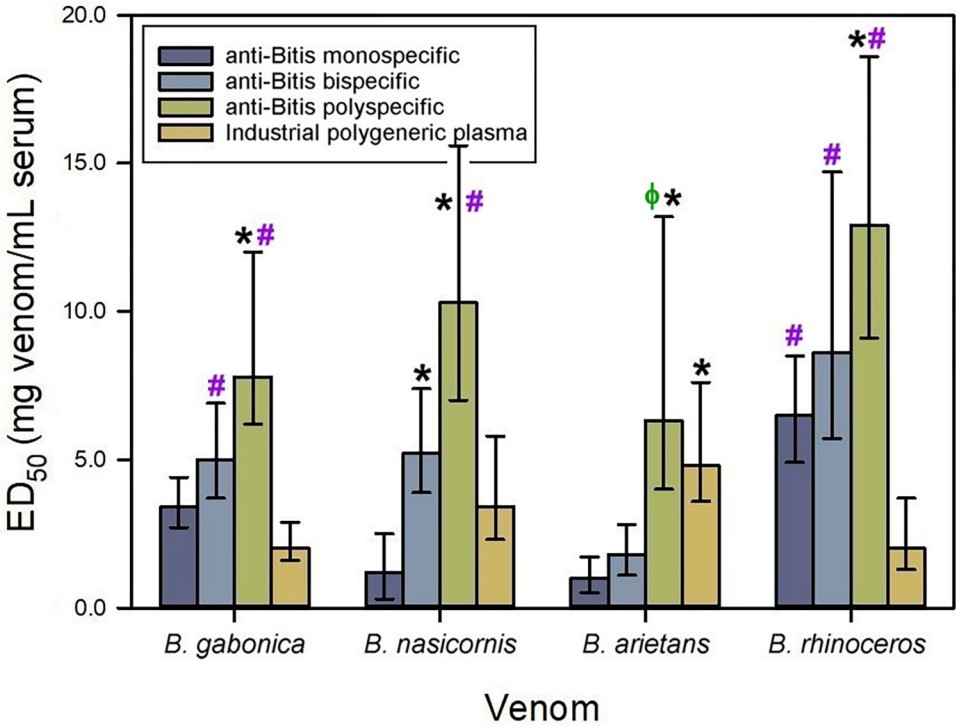

**Fig 2. Neutralization of the lethality induced by *Bitis* spp. venoms by equine anti-*Bitis* sera.** *B. gabonica* venom was used as immunogen to produce the monospecific antiserum; a mixture of equal parts of *B. gabonica* and *B. nasicornis* venoms were used to produce the bispecific antiserum; and a mixture of equal parts of *B. gabonica*, *B. nasicornis*, *B. arietans* and *B. rhinoceros* venoms were used to produce the polyspecific antiserum. Neutralization of lethality in mice is expressed as $ED_{50}$. Error bars represent the 95% confidence intervals. *Values significantly higher than the $ED_{50}$ of the monospecific serum. ϕ Values significantly higher than the $ED_{50}$ of the bispecific serum. #Values significantly higher than the $ED_{50}$ of the industrial polygeneric plasma (i.e., plasma obtained from horses chronically immunized with the venoms of *B. arietans*, *E. ocellatus*, *D. polylepis* and *N. nigricollis*).

The monospecific *Bitis* immunogen was enriched with the inclusion of *B. nasicornis* venom to have a bispecific immunogen composed of equal parts of venoms of *B. gabonica* and *B. nasicornis*. The use of this immunogen did not produce significant improvements when compared to the monospecific antiserum to neutralize *Bitis* venoms, except for *B. nasicornis* venom (Fig 2). This result is probably due to the antibody response induced by relevant toxins present in the venom of *B. nasicornis*, which are not shared with *B. gabonica* venom. In agreement with our findings, previous studies showed that antivenoms generated by immunization with the venoms of *B. arietans* [26], or *B. arietans* and *B. gabonica* [27], neutralized the venoms of *B. arietans*, *B. gabonica* and *B. rhinoceros*, but were much less effective in the neutralization of *B. nasicornis* venom.

The polyspecific *Bitis* immunogen was generated by mixing equal parts of *B. gabonica*, *B. nasicornis*, *B. arietans* and *B. rhinoceros* venoms. The use of this immunogen increased the neutralizing ability of the serum towards all the venoms in comparison to the monospecific immunogen (Fig 2), although no significant differences in the neutralization were observed when compared to the bispecific antiserum, except from the venom of *B. arietans*, which was neutralized to a higher extent by the polyspecific antiserum (Fig 2). Nevertheless, there is a trend of increase in the neutralizing potency of antisera associated with the use of a higher number of venoms in the immunizing mixture, probably because of the increment in the antigenic repertoire when using more venoms. Our findings are compatible with proteomic studies of *Bitis* sp venoms, which show interspecies variation in venom composition [6]. Determining the role of diversified and conserved antigens in the lethality of these venoms is a pending issue.

The neutralization of *B. arietans* venom was achieved to a higher extent when this venom was included in the immunizing mixture (polyspecific serum; Fig 2). This finding suggests that *B. arietans* venom has unique antigenic features, which in addition to the medical relevance and wide distribution of this species, justify their inclusion in the immunogens designed to produce antivenoms for sub-Saharan Africa.

Although there were not significant differences in the neutralizing ability between the bispecific and the polyspecific antisera against the majority of venoms, a general trend towards a higher neutralization by polyspecific antiserum was observed. Also, this antiserum showed a higher efficacy against the venom of *B. arietans*. Taken together, these observations suggest that the best option for neutralizing *Bitis* sp is to use the mixture of the four venoms for immunization.

When compared with the industrial polygeneric plasma (i.e., plasma obtained from horses chronically immunized with the venoms of *B. arietans*, *E. ocellatus*, *D. polylepis* and *N. nigricollis*), the monospecific serum had higher neutralizing ability towards the venom of *B. rhinoceros*; the bispecific serum had higher neutralizing ability towards the venoms of *B. gabonica* and *B. rhinoceros*; and the polyspecific serum had higher neutralizing ability towards all the venoms, except against *B. arietans* venom (Fig 2). These results suggest that the neutralization scope of the industrially produced EchiTAb-plus-ICP could be improved by the implementation of the polyspecific *Bitis* spp. immunogen.

## Neutralization of lethality by antisera raised against *Echis* sp venoms

The venom of *E. leucogaster* was used to generate the monospecific anti-*Echis* serum as per previous findings in a rabbit model [11]. As expected from this previous study, this antiserum neutralized the lethality induced by the venoms of *E. leucogaster*, *E. ocellatus* and *E. pyramidum*, albeit with different $ED_{50}$ values, showing a low efficacy against the venom of *E. ocellatus* (Fig 3).

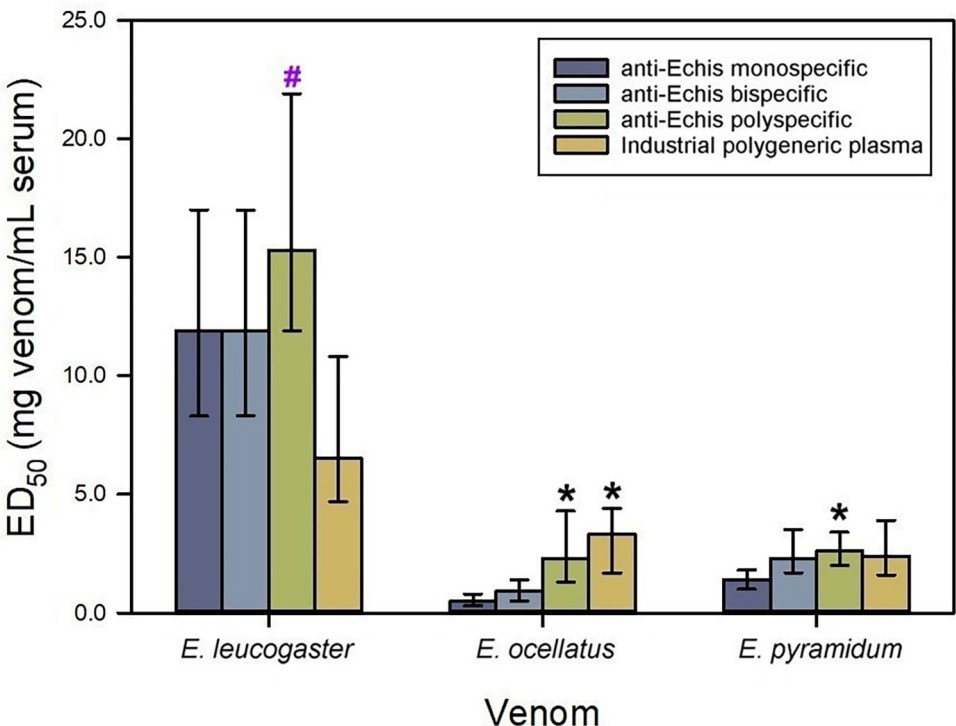

**Fig 3. Neutralization of the lethality induced by *Echis* spp. venoms by equine anti-*Echis* sera.** *E. leucogaster* venom was used as immunogen to produce the monospecific antiserum; a mixture of equal parts of *E. leucogaster* and *E. ocellatus* venoms were used to produce the bispecific antiserum; and a mixture of equal parts of *E. leucogaster*, *E. ocellatus* and *E. pyramidum* venoms were used to produce the polyspecific antiserum. Neutralization of lethality in mice is expressed as $ED_{50}$. Error bars represent the 95% confidence intervals. *Values significantly higher than the $ED_{50}$ of the monospecific serum. #Values significantly higher than the $ED_{50}$ of the industrial polygeneric plasma (i.e., plasma obtained from horses chronically immunized with the venoms of *B. arietans*, *E. ocellatus*, *D. polylepis* and *N. nigricollis*).

The inclusion of *E. ocellatus* venom to prepare the bispecific immunogen did not improve the neutralization of *E. ocellatus* venom (Fig 3), suggesting a similar immunogenic profile of the venoms of *E. leucogaster* and *E. ocellatus*. The subsequent addition of *E. pyramidum* venom to prepare the polyspecific immunogen, did not improve the neutralization of *E. leucogaster* venom. On the other hand, the polyspecific antiserum, but not the bispecific one, showed a higher neutralization of the venoms of *E. ocellatus* and *E. pyramidum* as compared to the monospecific antiserum (Fig 3), underscoring that the inclusion of *E. pyramidum* venom in the immunizing mixture enhances the scope of neutralization of the venoms of this genus. These results suggest that the polyspecific immunogen is the best option to produce anti-*Echis* serum with broad neutralization spectrum.

The industrial polygeneric plasma neutralized the venoms of *E. leucogaster*, *E. ocellatus* and *E. pyramidum* (Fig 3), which validates *E. ocellatus* venom as immunogen to induce antibody responses with broad neutralization scope within the genus. This agrees with previous observations on the neutralizing ability against several *Echis* sp venoms of EchiTAb-plus-ICP antivenom, which is generated from the industrial polygeneric plasma used in this study [26, 28]. Nonetheless, the fact that the polyspecific serum had higher ability to neutralize the venom of *E. leucogaster* than the industrial polygeneric plasma suggests that the use of this immunogen could enhance the overall neutralization scope of the EchiTAb-plus-ICP antivenom.

## Neutralization of lethality by antisera raised against *Dendroaspis* sp venoms

The monospecific *Dendroaspis* spp. immunogen was formulated with *D. jamesoni* venom, based on Gómez et al. (2024) [13]. In this case, the selection of this venom was not based on the neutralization of lethality in the rabbit study, because the efficacy of rabbit monospecific antisera was rather low; instead, we relied on immunochemical observations, i.e., ELISA and Western blot [13]. The antiserum generated in horses by using *D. jamesoni* venom neutralized the lethality of homologous and heterologous mamba venoms, except for *D. polylepis* (Fig 4), evidencing a partial intrageneric antigenic conservation.

The inclusion of *D. polylepis* venom in the bispecific immunogen resulted in a statistically significant increment of the neutralizing ability of the antiserum towards *D. polylepis* venom, but not towards the venoms of *D. jamesoni*, *D. viridis* or *D. angusticeps* (Fig 4). This result suggests the existence of unique antigens in the venom of *D. polylepis* which were not recognized by the antibodies induced by *D. jamesoni* venom. These observations agree with the proteomic analysis of *Dendroaspis* sp venoms, since the venom of *D. polylepis* differs from the others as it contains higher amounts of dendrotoxins and lower amounts of 3FTxs as compared to the other venoms of the genus [9].

The presence of *D. viridis* and *D. angusticeps* venoms in the polyspecific immunogen resulted in an improvement in the neutralization of *D. viridis* venom, as compared to the monospecific and bispecific antisera (Fig 4). When compared to the bispecific antiserum, no

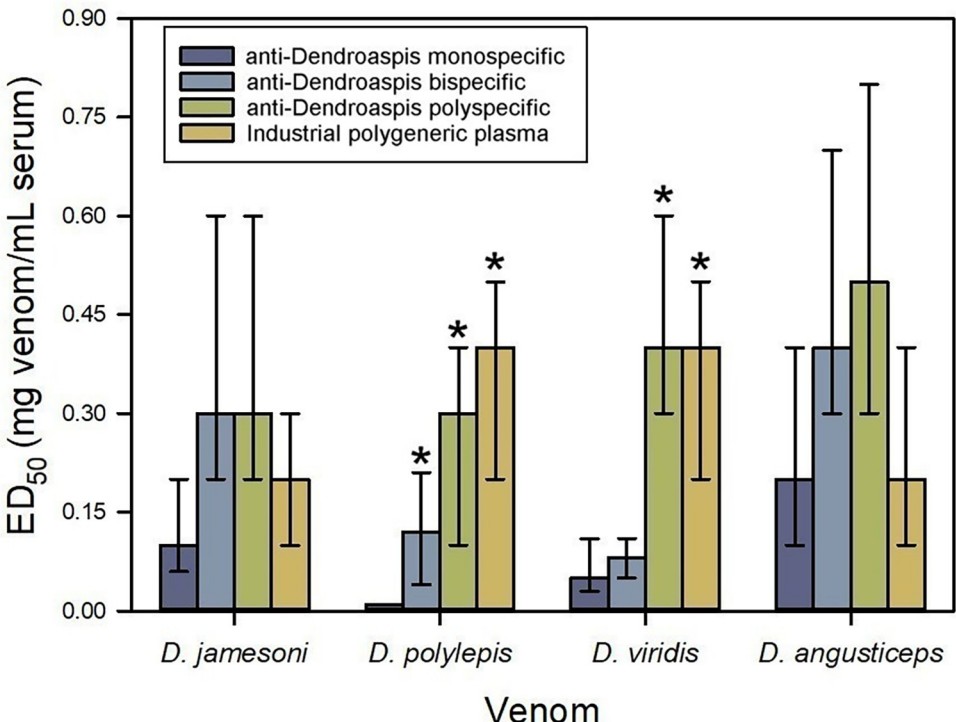

**Fig 4. Neutralization of the lethality induced by *Dendroaspis* spp. venoms by equine anti-*Dendroaspis* sera.** *D. jamesoni* venom was used as immunogen to produce the monospecific antiserum; a mixture of equal parts of *D. jamesoni* and *D. polylepis* venoms were used to produce the bispecific antiserum; and a mixture of equal parts of *D. jamesoni*, *D. polylepis*, *D. viridis* and *D. angusticeps* venoms were used to produce the polyspecific antiserum. Neutralization of lethality in mice is expressed as $ED_{50}$. Error bars represent the 95% confidence intervals. *Values significantly higher than the $ED_{50}$ of the monospecific serum.

improvement in neutralization was achieved by the polyspecific serum against the other three venoms tested (Fig 4). These findings suggest that the mixture of the four *Dendroaspis* sp venoms constitutes the best option for generating antivenoms of wide neutralizing coverage in the genus.

On the other hand, neither the monospecific, bispecific nor polyspecific sera had higher neutralizing ability than the industrial polygeneric plasma (Fig 4). This result could be due to a positive immunomodulation of the anti-*Dendroaspis* response by some of the co-immunogen venoms used to produce the polygeneric plasma (i.e., *B. arietans*, *E. ocellatus* or *N. nigricollis*). Positive immunomodulation was previously described in venoms of Latin American snakes during the industrial production of antivenom [29]. The existence of such immunologic interaction between African venoms must be experimentally demonstrated.

## Neutralization of lethality by antisera raised against spitting *Naja* sp venoms

The venom of *N. nigricollis* was used to formulate the monospecific immunogen of spitting (cytotoxic) *Naja* spp., following previous observations using rabbit antisera [12]. The antibody response induced by this venom was able to neutralize the lethality induced by all the venoms of spitting *Naja* assessed in this study, albeit with different $ED_{50}$s, having low efficacy against the venom of *N. katiensis* (Fig 5).

The bispecific immunogen (formulated with a mixture of equal parts of *N. nigricollis* and *N. katiensis* venoms), and the polyspecific immunogen (formulated with a mixture of equal parts of *N. nigricollis*, *N. katiensis*, *N. ashei*, *N. mossambica* and *N. nigricincta* venoms) did not exceed the ability of the monospecific immunogen to raise the production of neutralizing antibodies of the lethality induced by any of the spitting *Naja* spp. venoms.

These results underscore a high intrageneric conservation of antigens in the toxins responsible for lethality in mice in spitting *Naja* venoms, most likely 3FTx and $PLA_2$, the most abundant components in the venoms [30]. Consequently, an immunogen composed of just the venom of *N. nigricollis* is enough to formulate an immunogen suitable to produce anti-spitting *Naja* antiserum with broad neutralization spectrum.

When compared to the industrial polygeneric plasma, the monospecific antiserum had higher ability to neutralize *N. nigricollis* venom; the bispecific antiserum had higher ability to neutralize *N. nigricollis*, *N. ashei* and *N. mossambica* venoms; and the polyspecific antiserum was more effective in the neutralization of *N. nigricollis*, *N. ashei*, *N. mossambica* and *N. nigricincta* venoms (Fig 5). This result could be due to a negative immunomodulation of the anti-spitting *Naja* response by some of the co-immunogen venoms used to produce the polygeneric plasma (i.e., *B. arietans*, *E. ocellatus* or *N. nigricollis*). Negative immunomodulation was previously described in venoms of Latin American snakes during the industrial production of antivenom [31]. The existence of such immunologic interaction between African venoms must be experimentally demonstrated.

## Neutralization of lethality by antisera raised against non-spitting *Naja* sp venoms

The monospecific antiserum of non-spitting *Naja* spp. was generated by immunization with *N. senegalensis* venom, as per previous observations with antisera produced in rabbits [12]. The monospecific anti-non-spitting *Naja* antiserum was able to neutralize lethality of all neurotoxic *Naja* venoms, with variable $ED_{50}$s (Fig 6). The supplementation of the monospecific immunogen with *N. haje* venom resulted in a bispecific immunogen which was unable to increase the ability of the monospecific antiserum to neutralize the venoms (*N. haje* included),

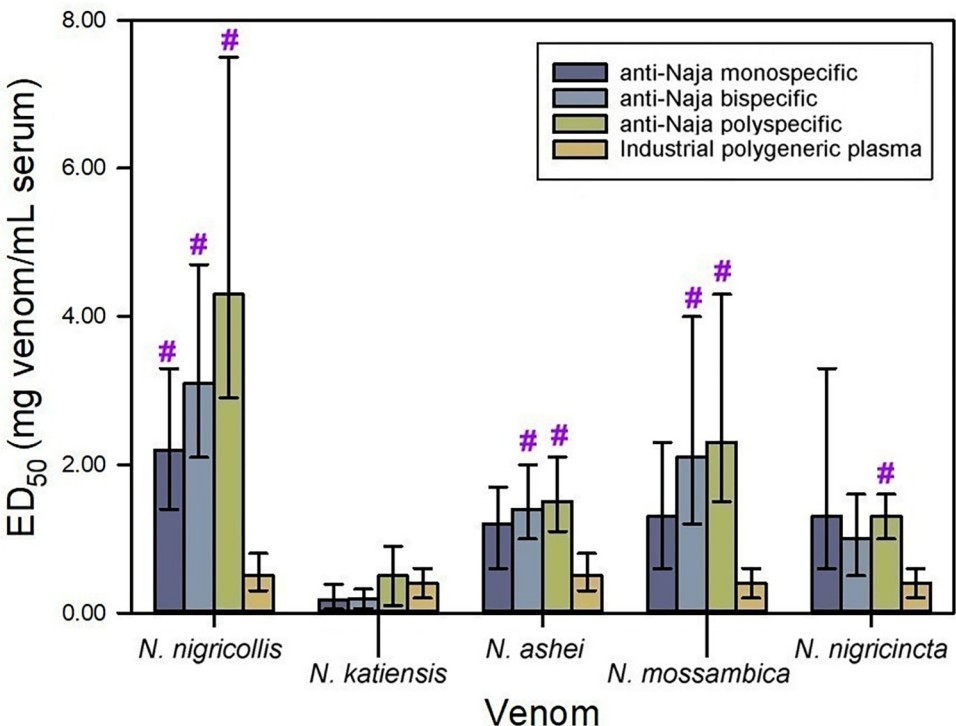

**Fig 5. Neutralization of the lethality induced by spitting *Naja* spp. venoms by equine anti-spitting *Naja* sera.** *N. nigricollis* venom was used as immunogen to produce the monospecific antiserum; a mixture of equal parts of *N. nigricollis* and *N. katiensis* venoms were used to produce the bispecific antiserum; and a mixture of equal parts of *N. nigricollis*, *N. katiensis*, *N. ashei*, *N. mossambica* and *N. nigricincta* venoms were used to produce the polyspecific antiserum. Neutralization of lethality in mice is expressed as $ED_{50}$. Error bars represent the 95% confidence intervals. #Values significantly higher than the $ED_{50}$ of the industrial polygeneric plasma (i.e., plasma obtained from horses chronically immunized with the venoms of *B. arietans*, *E. ocellatus*, *D. polylepis* and *N. nigricollis*).

except for that of *N. nivea* (Fig 6). This result suggests an antigenic similarity between *N. haje* and *N. nivea* venoms that is not shared by *N. senegalensis* venom.

The polyspecific immunogen formulated with a mixture of equal parts of *N. senegalensis*, *N. haje*, *N. anchietae*, *N. annulifera*, *N. melanoleuca* and *N. nivea* venoms had a similar performance as the bispecific immunogen. Polyspecific antiserum showed higher neutralizing efficacy, as compared to the monospecific one, against the venoms of *N. melanoleuca* and *N. nivea*. These results suggest the existence of unique antigens in relevant neurotoxins of *N. melanoleuca* and *N. nivea* venoms which are not shared with *N. senegalensis* venom. The venoms of neurotoxic cobras are characterized by having high amounts of neurotoxic 3FTxs [8, 10, 32].

No significant differences were observed between monospecific, bispecific and polyspecific antisera in the neutralization of the venoms of *N. senegalensis*, *N. haje*, *N. anchietae* and *N. annulifera*. However, when compared to the monospecific antiserum, the bispecific antiserum was more effective in the neutralization of *N. nivea* venom and the polyspecific antiserum was more effective in the neutralization of the venoms of *N. melanoleuca* and *N. nivea*. Thus, even though no significant differences were observed between the bispecific and the polyspecific antisera, the fact that the polyspecific one gave a higher neutralization against two venoms when compared to the monospecific antisera (Fig 6) suggests that the polyspecific venom mixture is the best option. Since our experimental design involved the addition of four venoms to

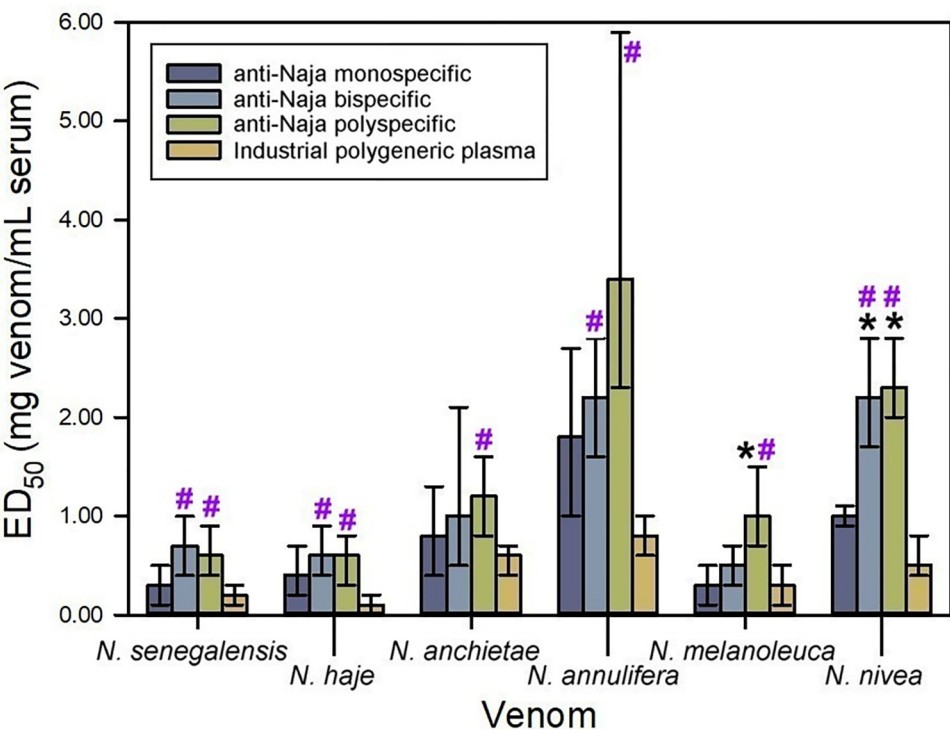

**Fig 6. Neutralization of the lethality induced by non-spitting *Naja* spp. venoms by equine anti-non-spitting *Naja* sera.** *N. senegalensis* venom was used as immunogen to produce the monospecific serum; a mixture of equal parts of *N. senegalensis* and *N. haje* venoms were used to produce the bispecific serum; and a mixture of equal parts of *N. senegalensis*, *N. haje*, *N. anchietae*, *N. annulifera*, *N. melanoleuca* and *N. nivea* venoms were used to produce the polyspecific serum. Neutralization of lethality in mice is expressed as $ED_{50}$. Error bars represent the 95% confidence intervals. *Values significantly higher than the $ED_{50}$ of the monospecific serum. #Values significantly higher than the $ED_{50}$ of the industrial polygeneric plasma (i.e., plasma obtained from horses chronically immunized with the venoms of *B. arietans*, *E. ocellatus*, *D. polylepis* and *N. nigricollis*).

the bispecific immunizing mixture, it is not known whether fewer venoms in the polyspecific mixture would suffice to neutralize all venoms, an issue that awaits further studies.

The neutralizing ability of the industrial polygeneric plasma was similar to that of the monospecific serum, with the exception of *N. nivea* which was neutralized to a higher extent by the monospecific antiserum. But it was surpassed by the bispecific serum in the neutralization of *N. senegalensis*, *N. haje*, *N. annulifera*, and *N. nivea* venoms; and by the polyspecific serum towards all the tested neurotoxic *Naja* venoms (Fig 6). These results are not surprising, because the immunizing mixture used to generate the industrial polygeneric plasma includes the venoms of *B. arietans*, *E. ocellatus*, *N. nigricollis* and *D. polylepis*, but none of the venoms of non-spitting *Naja* species. It is expected that the enrichment of the immunogen with venoms of non-spitting *Naja* should result in an improvement of the current neutralization scope of EchiTAb-plus-ICP.

## Implications of venom mixtures as immunogens

Our observations show that monospecific, bispecific and polyspecific immunogens of each genus were able to generate an immune response that neutralizes the lethality induced by the homologous venoms and, with a single exception, all the heterologous congeneric venoms (considering spitting and non-spitting *Naja* separately). In the case of monospecific antisera our findings generally agree with those obtained in a rabbit model [11–13], which highlighted

a high degree of immunological relatedness between venoms in each genus. These results could be explained by the conservation of antigenic characteristics in homologous toxins of phylogenetically related snakes. As a general trend, however, a higher neutralizing response was obtained with bispecific and, especially, with polyspecific antisera.

The variations observed between the neutralizing ability of the monospecific, bispecific and polyspecific sera, depending on the venom being neutralized, demonstrate the antigenic differences between congeneric venoms. Furthermore, the trend to increase the neutralizing ability of antisera as more venoms are included in the immunizing mixture highlights the antigenic differences between these venoms, both quantitative (i.e., the relative abundance of conserved antigens in different venoms) and qualitative (i.e., antigenic diversity in homologous toxins of different venoms).

The dilution of diversified antigens in the immunogen is proportional to the number of venoms included in the mixture. In contrast, conserved antigens are not diluted, regardless of how many venoms are included in the mixture. From the standpoint of manufacture of antivenoms, immunogenicity of conserved antigens allows the production of paraspecific formulations from immunogens composed of a reduced number of venoms. However, our findings suggest that, in the case of sub-Saharan African venoms, the inclusion of diversified antigens seems to increase the neutralization scope of the antisera, as evidenced by the neutralization of lethality. Thus, rather than decreasing the neutralization of some venoms due to the 'dilution' of venoms in a complex immunizing mixture, the antibody response and coverage of venoms seems to be enhanced by increasing the number of venoms in the mixture. This conclusion agrees with the concept that exposing the horse immune system to a diverse repertoire of toxin epitopes, by using a variety of venoms and isolated toxic fractions, generates antivenoms of wide neutralization scope [33, 34].

## Conclusions

Our findings provide novel information useful for the design of immunizing mixtures to generate antivenoms of broad neutralizing scope against the most relevant venoms of sub-Saharan African snakes. The antibody responses with the broadest intrageneric neutralizing scope were raised by the polyspecific immunogens of *Bitis* spp., *Echis* spp., *Dendroaspis* spp., and non-spitting *Naja* spp.; and the monospecific, bispecific and polyspecific immunogens of spitting *Naja* spp. None of the immunogens was associated with important adverse effects in the clinical or physical status of the horses. Except for the *E. pyramidum* venom, our experimental design does not allow to determine the contribution to the neutralizing scope made by each venom added to the bispecific immunogens to formulate the polyspecific immunogens. Testing the addition of each venom would involve the use of a large number of horses, which is against the principles of the 3Rs. Although this limitation does not affect the validity of the conclusions of this work, the evaluation of the contribution of each venom in the polygeneric immunogen remains as a pending task.

Our findings are based on the neutralization of lethal activity of venoms, the gold standard in antivenom efficacy assessment [3]. However, in the case of viperid venoms, it is relevant to additionally test the neutralization of other relevant toxic activities, such as hemorrhagic and coagulant/defibrinogenating effects. In the case of spitting *Naja* venoms, the neutralization of dermonecrotic activity is important because this is the main clinical manifestation of envenoming in humans [5]. In the case of non-spitting, neurotoxic *Naja* and mambas, the neutralization of lethality is sufficient to evaluate the preclinical efficacy of antivenoms. One limitation of the neutralization of lethality assay is that it has an intrinsic high variability, evidenced by the wide 95% CI, which precludes the detection of subtle differences in the neutralizing

capacity of antivenoms. Future work will focus on the development of experimental antivenoms generated by using mixtures of venoms of snakes from different genera in order to assess their immunomodulatory effects. In the long term, these efforts will contribute to the design of the most appropriate mixture of venoms for generating a pan-African antivenom of wide neutralizing scope.

## Supporting information

**S1 Table. Hematological and serum chemistry analyses of horses immunized with monogeneric immunogens composed of several venom mixtures. Values correspond to samples collected at the end of each immunization cycle (for monospecific, bispecific and polyspecific antisera).** * Internal reference values for adult creole horses of Instituto Clodomiro Picado. [1] Hematocrit values are expressed as percentage and correspond to the average ± SD (n = 4). [2] Hemoglobin values are expressed as g/dL and correspond to the average ± SD (n = 4). [3] CK: Creatine kinase. Values are expressed as IU/L and correspond to the average ± SD (n = 4). [4] Creatinine. Values are expressed as μmol/L and correspond to the average ± SD (n = 4). [5] Urea. Values are expressed as mg/dL and correspond to the average ± SD (n = 4). [6] AST: Aspartate transaminase. Values are expressed as IU/L and correspond to the average ± SD (n = 4). [7] ALP: Alkaline phosphatase. Values are expressed as IU/L and correspond to the average ± SD (n = 4). [8] GGT: Gamma-glutamyl transferase. Values are expressed as IU/L and correspond to the average ± SD (n = 4). [9] Total protein. Values are expressed as g/dL and correspond to the average ± SD (n = 4). [10] Albumin. Values are expressed as g/dL and correspond to the average ± SD (n = 4). [11] Gamma gap. Values are expressed as g/dL and correspond to the average ± SD (n = 4).
(DOC)

**S2 Table. Neutralization of the lethality induced in mice by sub-Saharan African venoms by equine monogeneric antisera*.** *Values correspond to the $ED_{50}$ and, in parentheses, the corresponding 95% CI. These results were used to construct Figs 2–6.
(DOC)

## Acknowledgments

The authors thank Christian Vargas, Jorge Gómez, Orlando Morales, and other colleagues at Instituto Clodomiro Picado for their technical support; and Andrés Hernández for the preparation of Fig 1. This work was performed in partial fulfillment of the doctoral degree of Andrés Sánchez at Universidad de Costa Rica.

## Author Contributions

**Conceptualization:** Andrés Sánchez, José María Gutiérrez, Guillermo León.

**Funding acquisition:** José María Gutiérrez, Guillermo León.

**Investigation:** Andrés Sánchez, Gina Durán, Álvaro Segura, María Herrera, Mariángela Vargas, Mauren Villalta, Mauricio Arguedas, Edwin Moscoso, Deibid Umaña, Aarón Gómez.

**Project administration:** José María Gutiérrez, Guillermo León.

**Writing – original draft:** Andrés Sánchez, José María Gutiérrez, Guillermo León.

**Writing – review & editing:** Andrés Sánchez, Gina Durán, Álvaro Segura, María Herrera, Mariángela Vargas, Mauren Villalta, Mauricio Arguedas, Edwin Moscoso, Deibid Umaña, Aarón Gómez, José María Gutiérrez, Guillermo León.

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
