## [Decision Letter · Decision Letter 0]

14 Mar 2024

Dear Dr. León,

Thank you very much for submitting your manuscript "Comparison of the intrageneric neutralization scope of monospecific, bispecific/monogeneric and polyspecific/monogeneric antisera raised in horses immunized with sub-Saharan African snake venoms" for consideration at PLOS Neglected Tropical Diseases. As with all papers reviewed by the journal, your manuscript was reviewed by members of the editorial board and by several independent reviewers. The reviewers appreciated the attention to an important topic. Based on the reviews, we are likely to accept this manuscript for publication, providing that you modify the manuscript according to the review recommendations. 

Specifically, your manuscript has been evaluated by three experts in the field, who unanimously recommended the publication after some minor changes. Please look at their comments and provide a response detailing how each topic was addressed (or not). Pay special attention to the recommendation of two of the reviewers about providing a diagram explaining the immunization scheme. I also appreciate the recommendation from the other reviewer about trying to make the article more educational and graphic to a wider audience and pointing out what makes these findings important.

Sincerely,

Inacio Loiola Meirelles Junqueira de Azevedo, Ph.D.

Academic Editor

Wuelton Monteiro

Section Editor

Reviewer's Responses to Questions

**Key Review Criteria Required for Acceptance?**

**Methods**

-Are the objectives of the study clearly articulated with a clear testable hypothesis stated?

-Is the study design appropriate to address the stated objectives?

-Is the population clearly described and appropriate for the hypothesis being tested?

-Is the sample size sufficient to ensure adequate power to address the hypothesis being tested?

-Were correct statistical analysis used to support conclusions?

-Are there concerns about ethical or regulatory requirements being met?

Reviewer #1: The methods are detailed and comprehensive. 

My main issue with this and the article is that while it is a masterclass for those who are familiar with the field, to be educational to a wider audience including those new to the field and policy makers and future funders for a neglected tropical disease there needs to be education. 

To that end, I would suggest the following for consideration: 

1. Diagram/Timeline of the inoculation procedure, including the different groups tested...While the methods are dense and detailed, they are hard to make a mental picture. 

2. Pictures of inoculation sites and how they are assessed 

3. There is a reference relating to adjuvants. Can the authors perhaps describe briefly their adjuvant compared to others such as those used in India? What is the advantage of the authors' adjuvant over say glycerine-based adjuvant used elsewhere?

Reviewer #2: Overall the study objectives are clear and appropriately designed. However in general the methods are a little too brief (heavily reliant on supplementary for key information that would be included in the main text. 

The methods section could also benefit from the following specific minor revisions;

> Animal management section - animal acclimatisation period? 

> More detail regarding the amount of venom given during immunisations. I know a full overview of this is given in supplementary, but this is basic important information I feel should be in the main text. 

> more detail regarding dosing/sampling regime. How long between immunisation dose and sample point etc. Line 207 says "different times" and would benefit from more detail. 

> Lines 2010-214; a little brief. when were the horses last immunised before plasma harvest for example? 

> ARRIVE guidelines; were mice randomly allocated? were experimenters blinded? gender of cd-1 mice?

> Line 246 - typo (2 DL instead of 2 LD)

Reviewer #3: The methodology used are appropriated and well conducted considering the objective of the manuscript. However, I suggest that the immunization scheme (Table 1) could be showed as the figure 1 to better comprehension of the results.

**Results**

-Does the analysis presented match the analysis plan?

-Are the results clearly and completely presented?

-Are the figures (Tables, Images) of sufficient quality for clarity?

Reviewer #1: No major concerns

Reviewer #2: Overall the results and data are clearly presented but could benefit from considering the following specific minor revisions;

> Lines 269-279 is not really results or discussion, and unnecessary if the methods are detailed enough. I suggest removing and ensuring the immunisation section within the methods is very clear. 

> Lines 280-283 – which injections of the 120 causes the lesions? All in primary imm with adjuvant? I think the actual imm schedule needs to be in the main manuscript. 

>Line 314; would be good to state in text the other bitis species that B. gabonica neutralised, rather than only having the information in the figure - this was done for the other genra examined. 

> Lines 336-341 – the sentence starts comparing poly to mono, then comments that there were no significant differences between what I presume is poly and bi except for b. arietans – however these stats aren’t described in figure legends (* = vs mono, # = vs industrial). Suggest having a marker for different to bispecific if referring to that comparison in the text (example line 350-352 this exact comparison is remarked on). This should also be applied throughout the manuscript where relevant. 

> Line 354-355 typo/bad gramma, “is the using the”.

> Lines 354-355 – the data isn’t really there to support all 4 – the neutralisation of B. rhinoceros was already good with both mono and bi, you could just include the b. ari to improve recognition against that species and stick at 3. But I take the point that it probably doesn’t matter if you include a 4th. It is a minor weakness of the study that it jumps from mono to bi to 4+ for everything except echis as it makes it harder to define the truly important ones in and conclusively say all or just some are needed. But this is more ethical from a 3Rs standpoint. 

> Line 385-390 - It is interesting that bispecific isn’t sig better than mono for e. oce considering that was what was added in to the immunisation mix. A comment on this with reference to known literature regarding e. oce would improve the discussion element of this section. 

> Line 436-439 – I dont think the data supports the suggestion that all 4 constitute the best option. I would argue that d. viridis is not needed, this was as effectively neutralised by the industrial blend indicating den pol in the mix is enough, and that the mix of den jam and den pol is enough to neutralise d. angusticeps. 

> Line 439-440; Whilst not significantly, there are trends towards this for Den jam and D. ang, and previous discussion for other genra have made use of trends for support of opposing points (line 350-352 for example). Just check use of trends vs significantly different arguments for consistency in supporting data statements. 

> Line 526-527 – there is a large difference between 2 and 6 venoms; I think a comment that the current experimental design makes it hard to infer which no spitting species being added would or wouldn’t increase overall neutralising ability is warranted. Saying bi or poly is quite generic and doesn’t add value to future study designs. Especially as 536-538 suggests adding these would benefit an existing manufactured product - would the authors add just senegalensis and haje, or more? 

> figure legends don't refer to mice at all

As a general comment - I think the discussion lacks a bit of depth throughout; it is a very observational discussion. There are some wider discussion points (positive and negative immunomodulation etc), but further venomic data etc could be added throughout to aid discussion and interpretation of results. 

Supplemental 

> Supp 1 - n numbers missing from legend

> Supp 3 table title too brief. Neutralisation of lethality in what? The table alone doesn’t reflect that is it a murine model

Reviewer #3: Yes, the results are appropriated presented, however some information should be add to the discussion.

**Conclusions**

-Are the conclusions supported by the data presented?

-Are the limitations of analysis clearly described?

-Do the authors discuss how these data can be helpful to advance our understanding of the topic under study?

-Is public health relevance addressed?

Reviewer #1: No major concerns

Reviewer #2: I think overall the observations and subsequent conclusions are well supported with exception to those noted in the results comments above. The limitations are clearly described, although comment could be given to the study design of mono to bi to varying number of species for the poly blend and the difficulties this can cause in drawing conclusions. There is clear evidence for how the work can further understanding of the topic.

Reviewer #3: Appropriated.

**Summary and General Comments**

Reviewer #1: Very comprehensive work.

Reviewer #2: Authors detail the method for producing antivenom, the medically relevant genera in sSA and the pathology incurred by each, the medically relevant toxins driving pathology, touching on the differences in immunogenicity and host immune response. They refer to a previous rabbit model and detailed species from each genera that had broad intrageneric neutralisation when used as immunogens. Their study uses a mix of venoms from bitis, echis, dendroaspis and naja spp in a scaled up model in horses and look at intrageneric neutralisation of lethality. They clearly demonstrate the changes that including further species within an immunisation mix can cause, and offer opinion on which species of a genre may supplement and enhance existing immunisation methods. The successful translation of work from a previous rabbit study in the same group to a relevant manufacturing model is important. A common thought in the field is that increasing the number of venoms in an immunisation mix to broaden intrageneic neutralisation dilutes the relevant pool of neutralising antibodies against a specific species, and this work provides good evidence that this is not the case.

Reviewer #3: The aim of this manuscript is to contribute with the improvement in the sera production in horses with high ability to neutralize the envenomation by sub-Saharan African snake bites. For this, the authors used an experimental approach for obtaining distinct antivenoms after a scheme of immunization of horses with snake venom combinations according to monospecific, bispecific/monogeneric and polyspecific/monogeneric specificity. The results are consistent showing that the immunization protocols did not induce systemic effect with grade 3 of local lesions in the horses. Furthermore, the data evaluating the capability of distinct antisera to neutralize the lethal activity of the venoms are very interesting. However, to improve the manuscript and understanding the results, some points should be considered and discussed for the publication. 

- The authors should add more information about the industrial polygeneric plasma used in the experiments considering the kinetic of specific-antibody production and the decline of specific-antibody production. 

- The immunization scheme (Table 1) could be showed as the figure 1 to better comprehension of the results.

- In general, there is a correlation between the magnitude/affinity the antibody production against antigens/immunogens and the antigenic boosters used to obtain the antisera, how the authors discuss this considering that the time-point of the bleeding to obtain the monospecific, bispecific and polyspecific antisera for comparison the neutralizing potential of each one? The authors have more information or results about this?

- More information about the diversity and immunogenicity between the venoms of Bitis spp to produce the antisera to reinforce the results and “the increment in the antgenic repertoire when using more venoms”, (lines 343-344, page 14) and also the relevance to add the B.arietans venom to the immunizing mixture to improve the neutralizing ability of the polyspecifc antisera (lines-345-39, page 14), as suggested by the authors in the text.

PLOS authors have the option to publish the peer review history of their article (what does this mean?). If published, this will include your full peer review and any attached files.

Reviewer #1: No

Reviewer #2: No

Reviewer #3: No

**Editorial and Data Presentation Modifications?**

Reviewer #1: I would like to see this article focus on educating new readers. This will get published as is, but to be successful it should teach a broader swath of readers. Especially considering the big press on new approaches and some hype about new antibody formats...this communication is critically important. What makes these findings important, interesting and how do they apply to the future of the field and address global shortage?

Reviewer #2: (No Response)

Figure Files:

While revising your submission, please upload your figure files to the Preflight Analysis and Conversion Engine (PACE) digital diagnostic tool, https://pacev2.apexcovantage.com. PACE helps ensure that figures meet PLOS requirements. To use PACE, you must first register as a user. Then, login and navigate to the UPLOAD tab, where you will find detail

---

## [Decision Letter · Decision Letter 1]

2 May 2024

Dear Dr. León,

We are pleased to inform you that your manuscript 'Comparison of the intrageneric neutralization scope of monospecific, bispecific/monogeneric and polyspecific/monogeneric antisera raised in horses immunized with sub-Saharan African snake venoms' has been provisionally accepted for publication in PLOS Neglected Tropical Diseases.

Best regards,

Inacio Junqueira de Azevedo, Ph.D.

Academic Editor

Wuelton Monteiro

Section Editor

Reviewer's Responses to Questions

**Key Review Criteria Required for Acceptance?**

**Methods**

-Are the objectives of the study clearly articulated with a clear testable hypothesis stated?

-Is the study design appropriate to address the stated objectives?

-Is the population clearly described and appropriate for the hypothesis being tested?

-Is the sample size sufficient to ensure adequate power to address the hypothesis being tested?

-Were correct statistical analysis used to support conclusions?

-Are there concerns about ethical or regulatory requirements being met?

Reviewer #1: I have no major concerns remaining.

Reviewer #2: Methods have been substantially improved by the extra details requested by the reviewers.

**Results**

-Does the analysis presented match the analysis plan?

-Are the results clearly and completely presented?

-Are the figures (Tables, Images) of sufficient quality for clarity?

Reviewer #1: No major concerns.

Reviewer #2: all comments addressed well by the authors, the quality of the discussion of results has been improved.

**Conclusions**

-Are the conclusions supported by the data presented?

-Are the limitations of analysis clearly described?

-Do the authors discuss how these data can be helpful to advance our understanding of the topic under study?

-Is public health relevance addressed?

Reviewer #1: No major concerns. I think the authors have done an admirable job making this work accessible at different levels of need/education.

Reviewer #2: the limitations of the study have been expanded, authors have addressed all comments well.

**Editorial and Data Presentation Modifications?**

Reviewer #1: No major concerns. I think the authors have done an admirable job making this work accessible at different levels of need/education.

Reviewer #2: Table one could do with expanding on to make it a stand alone table - for example it has venom dose in brackets but this isn't described in the title or table itself.

**Summary and General Comments**

Reviewer #1: No major concerns. I think the authors have done an admirable job making this work accessible at different levels of need/education.

Very solid piece of work with relevance to the field academically and pragmatically.

Reviewer #2: all comments addressed well and happy to recommend for publication.

PLOS authors have the option to publish the peer review history of their article (what does this mean?). If published, this will include your full peer review and any attached files.

Reviewer #1: No

Reviewer #2: No

---

## [Editor Report · Acceptance letter]

13 May 2024

Dear Dr. León,

We are delighted to inform you that your manuscript, "Comparison of the intrageneric neutralization scope of monospecific, bispecific/monogeneric and polyspecific/monogeneric antisera raised in horses immunized with sub-Saharan African snake venoms," has been formally accepted for publication in PLOS Neglected Tropical Diseases.

Best regards,

Shaden Kamhawi

co-Editor-in-Chief

Paul Brindley

co-Editor-in-Chief
